# Risk-Driven Design of Perception Systems

**Anthony L. Corso** *
Department of Aeronautics and Astronautics
Stanford University
Stanford, CA
acorso@stanford.edu

**Sydney M. Katz** *
Department of Aeronautics and Astronautics
Stanford University
Stanford, CA
smkatz@stanford.edu

**Craig Innes**
School of Informatics
University of Edinburgh
Edinburgh, UK
craig.innes@ed.ac.uk

**Xin Du**
School of Informatics
University of Edinburgh
Edinburgh, UK
x.du@ed.ac.uk

**Subramanian Ramamoorthy**
School of Informatics
University of Edinburgh
Edinburgh, UK
s.ramamoorthy@ed.ac.uk

**Mykel J. Kochenderfer**
Department of Aeronautics and Astronautics
Stanford University
Stanford, CA
mykel@stanford.edu

## Abstract

Modern autonomous systems rely on perception modules to process complex sensor measurements into state estimates. These estimates are then passed to a controller, which uses them to make safety-critical decisions. It is therefore important that we design perception systems to minimize errors that reduce the overall safety of the system. We develop a risk-driven approach to designing perception systems that accounts for the effect of perceptual errors on the performance of the fully-integrated, closed-loop system. We formulate a risk function to quantify the effect of a given perceptual error on overall safety, and show how we can use it to design safer perception systems by including a risk-dependent term in the loss function and generating training data in risk-sensitive regions. We evaluate our techniques on a realistic vision-based aircraft detect and avoid application and show that risk-driven design reduces collision risk by 37 % over a baseline system.

## 1   Introduction

The design of reliable perception systems is a key challenge in the development of safety-critical autonomous systems [1], [2]. Modern perception systems are often required to predict state information from complex, high-dimensional inputs such as images or LiDAR data [3]–[5]. This information is then passed to a controller, which uses the state estimate to make safety-critical decisions. For example, vision-based perception systems have been proposed for detect and avoid applications in aviation [5]. It is important that perception systems produce accurate estimates, and these systems are typically trained to minimize overall perceptual error using a regression loss on a set of training data. This training method, however, fails to account for the effect of perceptual errors on the performance of the fully-integrated, closed-loop control system. In particular, not all errors will have an equal

---

*equal contribution

36th Conference on Neural Information Processing Systems (NeurIPS 2022).

effect on overall performance. While some errors pose minimal risk, others may lead to catastrophic outcomes [6], [7].

**Contribution** In this work, we develop a technique to design perception systems that are sensitive to the overall risk of the closed-loop system in which they operate. We formulate a risk function that quantifies the downstream effect on safety of making a given perceptual error in a given state. We then show how the risk function can be used to design safer perception systems by incorporating it into the loss function during training and using it to focus data collection efforts on the most risk-sensitive regions of the state space. Finally, we analyze the impact of our risk-driven design approach on the safety of a realistic vision-based aircraft detect and avoid system. We show that our approach is able to reduce collision risk by up to 37 % over a baseline perception system.

## 2 Background

This work uses a notion of conditional value at risk (CVaR) and a Markov decision process (MDP) formulation to estimate the effect of perceptual errors on the safety of a closed-loop system. This section details the necessary background on these topics.

**Conditional Value at Risk (CVaR)** Let $X$ be a bounded random variable and $F(x) = P(X \leq x)$ be its cumulative distribution function. The value at risk (VaR) is the highest value that $X$ is guaranteed not to exceed with probability $\alpha$ written as

$$\text{VaR}_\alpha(X) = \min\{x \mid F(x) \geq \alpha\} \tag{1}$$

Using the VaR, we can derive the CVaR for a given $\alpha \in (0, 1)$, which represents the expected value of the top $1 - \alpha$ quantile of the probability distribution over $X$. More formally,

$$\text{CVaR}_\alpha(X) = \mathbb{E}[X \mid X \geq \text{VaR}_\alpha(X)] \tag{2}$$

If $X$ is random variable that represents cost, $\text{CVaR}_\alpha(X)$ will correspond to the expected value of the worst $\alpha$-fraction of costs. As $\alpha$ approaches 1, $\text{CVaR}_\alpha(X)$ approaches the cost of the worst-case outcome. As $\alpha$ approaches 0, $\text{CVaR}_\alpha(X)$ approaches the expected value of $X$. CVaR has some desirable mathematical properties [8] that have often made it the preferred risk metric in risk-sensitive domains such as finance [9]–[11] and robotics [8].

**Markov Decision Process (MDP)** An MDP is a way of encoding a sequential decision making problem in which an agent's action at each time step depends only on its current state [12]. An MDP is defined by the tuple $(S, A, T, C, \gamma)$, where $S$ is the state space, $A$ is the action space, $T(s' \mid s, a)$ is the probability of transitioning to state $s'$ given that we are in state $s$ and take action $a$, $C(s, a)$ is the cost of taking action $a$ in state $s$, and $\gamma$ is the discount factor. A stochastic policy for an MDP, written as $\pi(a \mid s)$, denotes the likelihood of taking action $a$ in state $s$ and induces a distribution over future costs. We define a cost distribution function $Z^\pi(s, a)$ that maps a given state $s$ and action $a$ to a random variable representing the sum of discounted future costs obtained by taking action $a$ from state $s$ and subsequently following policy $\pi$ as follows

$$Z^\pi(s_t, a_t) = C(s_t, a_t) + \sum_{t'=t+1}^{\infty} \gamma^{t'} C(s_{t'}, a_{t'}) \tag{3}$$

where $s_{t'+1} \sim T(\cdot \mid s_{t'}, a_{t'})$ and $a_{t'} \sim \pi(\cdot \mid s_{t'})$. To evaluate a given policy $\pi$, we calculate a state-action value function, denoted $Q^\pi(s, a)$, as the expected sum of discounted future costs

$$Q^\pi(s, a) = \mathbb{E}\left[Z^\pi(s, a)\right] \tag{4}$$

In traditional reinforcement learning, $Q^\pi$ is the optimization objective [12]; however, in safety-critical domains we may wish evaluate a policy based on an upper quantile of the worst-case outcomes. In this case, rather than calculating the expectation of future costs, we define a new state-action value function $Q_\alpha^\pi(s, a)$ that represents the CVaR of the distribution of future costs as

$$Q_\alpha^\pi(s, a) = \text{CVaR}_\alpha[Z^\pi(s, a)] \tag{5}$$

Intuitively, $Q_\alpha^\pi(s, a)$ is the expected value of the $\alpha$-percentile of worst-case outcomes. We note that when $\alpha = 0$, eq. (5) reduces to eq. (4). One way to solve for $Q_\alpha^\pi(s, a)$ is to first solve for the cost distribution function $Z^\pi(s, a)$, which can be done using techniques from distributional reinforcement learning [13]. The CVaR can then be computed using this distribution. Other techniques that do not require explicitly solving for the distribution over costs may also be applied [10], [14].

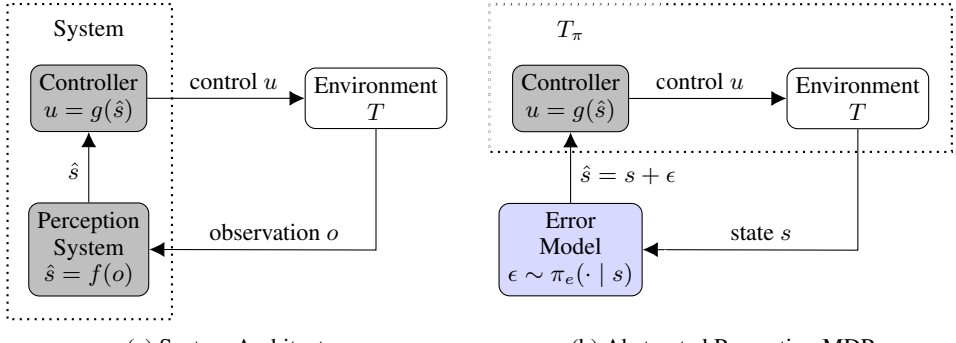

(a) System Architecture            (b) Abstracted Perception MDP

Figure 1: Problem formulation.

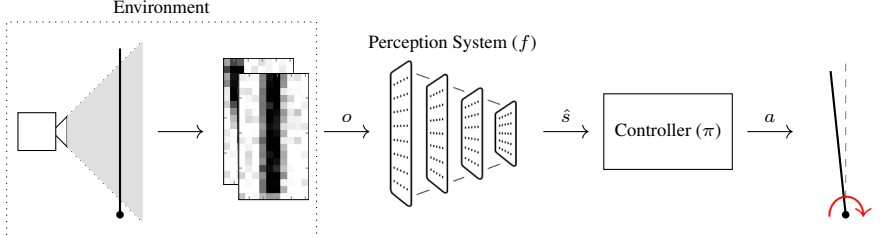

Figure 2: Overview of the inverted pendulum example system.

## 3 Approach

We consider the setting shown in fig. 1a, in which a system is composed of a perception module $f$ and a controller $g$. Given an observation $o \in \mathcal{O}$ of the true state $s \in \mathcal{S}$, the perception system produces an estimate of the state $\hat{s} = f(o)$. The controller produces a control input $u = g(\hat{s})$ and the system transitions to state $s'$ with probability $T(s' \mid s, u)$. We assume that the controller $g$ is given, and we wish to design a perception module $f$ such that the overall system satisfies a safety property.

**Pendulum example**   We will use a vision-based inverted pendulum problem as a running example to describe our approach (see Figure 2). We assume that a camera produces noisy image observations of the pendulum's current state $s = [\theta, \omega]$ where $\theta$ is its angle from the vertical and $\omega$ is its angular velocity. To estimate both position and velocity information from the images, we use two consecutive image frames for each observation. The perception system is a multi-layer perceptron (MLP) that takes in the image observations and produces an estimate of the state. Using the state estimate, the controller selects a torque to keep the pendulum upright. We specify a safety property that requires that $|\theta| < \pi/4$ at all time steps. We use a rule-based controller that satisfies the safety property under perfect perception; however, image noise, image downsampling, and the finite capacity of the MLP make perception errors inevitable, and the goal is to design a perception system that limits the occurrence of high risk errors. Appendix A.1 summarizes the inverted pendulum model.

### 3.1 Overview

The key insight in our approach is to abstract the perception system using a notional model of perception errors and analyze the effect of specific perception errors on the performance of the closed-loop system. As shown in fig. 1b, we model the errors $\epsilon \in \mathcal{E}$ of the abstracted perception system as a stochastic policy $\pi_e(\epsilon \mid s)$ in an MDP. We note that this representation implicitly takes into account the possible observations for state $s$ by directly outputting the errors the perception system could make when processing each observation. The cost function of the MDP is chosen to reflect the risk of a being in a state with respect to a safety property. By formulating the problem in this way, we can use the policy evaluation techniques outlined in section 2 to evaluate the risk of making perception error $\epsilon$ in state $s$. Given that $\pi_e$ is an abstraction of the perception system, it will not exactly reflect the true distribution of perception errors. To account for this mismatch, we are

conservative in our risk estimate by evaluating the policy according to eq. (5). Once computed, the risk function represents the downstream impact of making a given perception error in the current state. This function can therefore be used in a supervised learning setting to encourage a model to avoid making high-risk errors. We also show that the risk function can be used to identify error-sensitive regions of the state space, from which additional training data can be collected.

## 3.2 Risk Function

Our goal is to define a risk function $\rho(s, \epsilon)$ that quantifies the risk of making perception error $\epsilon$ in state $s$ when using a controller $g$. We model the perception error as a stochastic policy $\pi_e(\epsilon \mid s)$ that reflects a notional distribution over errors in state $s$. The controller produces an action based on the perceived state $\hat{s}$, which is computed from $s$ and $\epsilon$. For additive noise, the resulting transition function for the abstracted perception MDP is

$$T_\pi(s' \mid s, \epsilon) = T(s' \mid s, g(s + \epsilon)) \tag{6}$$

The error model $\pi_e$ can be designed to match the performance of a previously trained baseline perception system or using domain knowledge. If we are confident that our choice closely matches the errors we will see when designing the perception system, we can set our risk function to match the state action-value function in eq. (4). However, if we have not yet designed the perception system, there will be a mismatch between our estimate of $\pi_e$ and the true distribution of perception errors.

In these scenarios, we may want a more conservative risk function in which we consider the CVaR of the distribution over future costs when making perception error $\epsilon$ in state $s$ given that all future perception errors are distributed according to $\pi_e$. Therefore, we select our risk function as follows

$$\rho_\alpha^{\pi_e}(s, \epsilon) = Q_\alpha^{\pi_e}(s, \epsilon) \tag{7}$$

where $Q_\alpha^{\pi_e}(s, \epsilon)$ refers to the CVaR state-action value function defined in eq. (5) for the abstracted perception MDP. The parameter $\alpha$ smoothly controls the degree of conservatism. When $\alpha = 0$, the risk function is equivalent to the expected cost when following the notional perception model. When $\alpha = 1$, we consider only the worst-case sequence of possible perception errors.

To solve for the CVaR state-action value function we apply distributional dynamic programming to compute the distribution over costs. Distributional dynamic programming (alg. 5.3 in Bellemare *et al.* [13]) resembles value iteration where the Bellman operator is replaced with a distributional version to update a parameterized distribution. In this work, we approximate the distribution over costs using a categorical distribution with a fixed number of discrete costs. Additionally, we discretize the states and actions and use a linear weighting function for local approximation. To extend our method to problems with larger state and action spaces, we could apply function approximation techniques developed in prior work [15]–[17], but we leave this as future work. Once we solve for the cost distributions, we can calculate $\rho_\alpha^{\pi_e}(s, \epsilon)$ for any value of $\alpha$ using eq. (2) with minimal additional computation.

**Pendulum example** For the inverted pendulum system, we model the perception errors $\epsilon = [\epsilon_\theta, \epsilon_\omega]$ as additive noise on the true state ($\hat{s} = s + \epsilon$) distributed according to a multivariate Gaussian distribution with zero mean and diagonal covariance. The cost function penalizes the absolute deviation from the vertical $|\theta|$. See appendix A.2 for more detail. Figure 3 shows the values of the pendulum risk function at the state $s = [0.2, 0.0]$ over the set of errors for multiple values of $\alpha$. A higher value of $\alpha$ will put more weight on worst-case perception errors in the future, so as $\alpha$ increases, the overall risk values increase. The risk function also provides a notion of *which* perception errors are most risky in a given state. For example, the optimal action when $s = [0.2, 0.0]$ is to apply a negative torque to move the pendulum closer to upright, but if perception errors are negative, the resulting $\hat{s}$ may indicate that the pendulum is at a smaller angle and already moving in the negative direction as shown in the leftmost image on fig. 3. This estimate may cause the controller to produce a smaller torque or even a torque in the positive direction, leading to high risk of falling over.

## 3.3 Risk-Sensitive Loss Function

Perceptual errors are inevitable whenever there is sensor noise, occlusions, limited training data, or limited model capacity. Our goal is to ensure that the errors that are made by the perception system are in the direction of lower, rather than higher, risk. We incorporate this notion into our design by

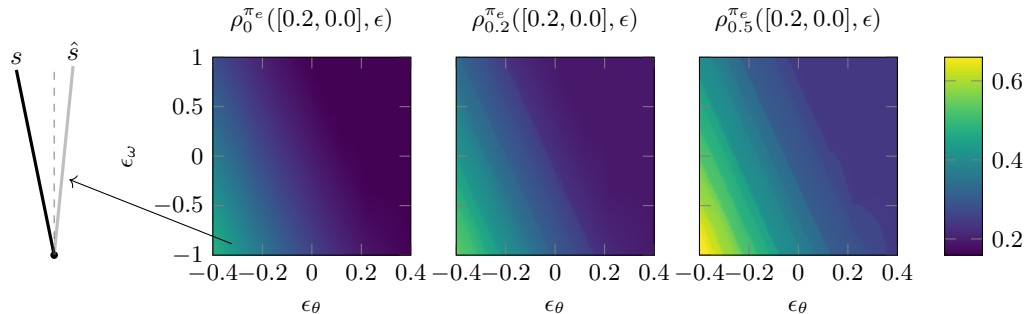

Figure 3: Values of $\rho_\alpha^{\pi_e}([0.2, 0.0], \epsilon)$ for different values of $\alpha$. The state and an example error are depicted on the left.

formulating a risk-sensitive loss function. Let $\mathcal{L}(s, \hat{s})$ be the standard loss function for the baseline perception system such as mean squared error for regression tasks or cross entropy for classification tasks. We define a risk-sensitive loss function for a single datapoint with additive noise as follows

$$\mathcal{L}_R(s, \hat{s}) = \mathcal{L}(s, \hat{s}) + \lambda \rho_\alpha^{\pi_e}(s, \hat{s} - s) \tag{8}$$

where $\lambda$ is a hyperparameter that controls the relative weighting of each term. The first term in the loss function pushes all perception errors toward zero, while the second term seeks to minimize the risk of the current errors that the model makes. We note that this is different from simply minimizing perception errors in states with high risk values. If the risk function outputs the same value for a given state regardless of perception error, the gradient of the risk term in the loss function will be zero.

**Pendulum example** For the inverted pendulum problem, we use a mean squared error baseline loss function, which penalizes positive and negative errors equally. However, a positive and negative error of the same magnitude do not necessarily have an equal effect on the overall risk of the closed-loop system. For example, consider the results shown in fig. 3 where the pendulum is positioned at an angle of 0.2 from the vertical. The risk function shows that a perception error of $\epsilon_\theta = 0.2$ ($\hat{\theta} = 0.4$) is less risky than a perception error of $\epsilon_\theta = -0.2$ ($\hat{\theta} = 0.0$). Intuitively, predicting that the pendulum is tipped further in the positive direction will still result in a negative torque to move the pendulum closer to upright. In contrast, predicting that the pendulum is perfectly upright will cause the controller to output zero torque. The second term in eq. (8) accounts for this effect.

### 3.4  Risk-Driven Data Generation

In addition to considering the directionality of the errors, we can use the risk function we formulated in section 3.2 to identify states that are especially sensitive to perception errors and generate more training data in those regions. In effect, this process gives higher weight to risky states during training. Since training with a regression loss will push all errors toward zero, we define risky states as states in which making a nonzero perception error will result in a high risk value compared to the risk of zero error. We use a weighting function $w_\alpha(s)$ to capture this notion as follows

$$w_\alpha(s) = \max_{\epsilon \in \mathcal{E}} \rho_\alpha^{\pi_e}(s, \epsilon) - \rho_\alpha^{\pi_e}(s, 0) \tag{9}$$

We note that $w_\alpha(s)$ is not the same as (and not necessarily correlated with) the risk of being in state $s$. Instead, $w_\alpha(s)$ represents the risk of making a nonzero perception error in state $s$. For instance, states in which failure is inevitable regardless of perception error will have high risk values but low weight assigned to them according to $w_\alpha(s)$. To sample data, we use rejection sampling with $w_\alpha(s)$ as the likelihood function.

**Pendulum example** Figure 4 illustrates each term in eq. (9) for the inverted pendulum example problem. The lower left and upper right regions of the state space represent states in which the pendulum is at a high angle and moving in a direction away from the vertical. Due to underactuation of the pendulum system, the states at these extremes are guaranteed to lead to failure regardless of perception error. States in the central band of each plot, on the other hand, constitute states in which the pendulum is likely to remain upright as long as proper control is applied. This band widens when we consider the effect of having zero perception error compared to the worst-case perception

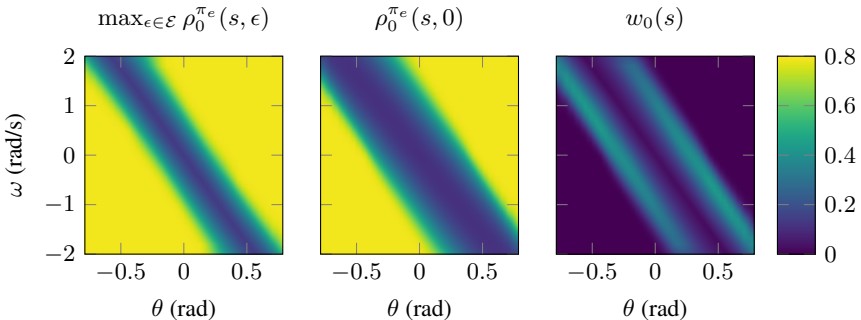

Figure 4: Illustration of eq. (9).

error. By subtracting these quantities, we can understand in which states having zero perception error is most critical. Given a limited data budget, we can focus our data collection on these states by sampling data according to $w_\alpha(s)$.

### 3.5 Hypotheses

We design the experiments in this work to test the following two hypotheses:

- **H1**: A perception module trained using the risk-sensitive loss function proposed in section 3.3 will result in a safer closed-loop system than one trained using the baseline loss on the same dataset.
- **H2**: A perception system trained on data sampled according to the risk-driven data generation technique proposed in section 3.4 will result in a safer closed-loop system than one trained on data uniformly sampled throughout the state space.

**Pendulum example** We evaluate the inverted pendulum by measuring the mean time to failure over 100 trajectories each with 500 time steps from random initial states. We report the mean and standard error of this metric over 5 trials with 500 being the maximum attainable value. To test **H1**, we train a baseline perception system using a mean squared error loss function and a risk-driven perception system using the loss function described in section 3.3 on 10,000 uniformly sampled data points. To make the perception problem more challenging, we add noise to the images. The risk sensitive loss function improved the mean time to failure from the baseline of $234 \pm 62$ to $492 \pm 6$. To test **H2** on the inverted pendulum system, we limit the perception system to a data budget of 50 training examples. We train a baseline perception system on data sampled uniformly throughout the state space and a risk-driven perception system on data sampled according to the weighting function shown in fig. 4. The mean time to failure for the baseline system is $152 \pm 89$, while the time to failure for the system trained of the risk-driven data was 500 for all trials. See appendix A.3 for full results.

## 4 Vision-Based Detect and Avoid Application

Aircraft collision avoidance systems use information gathered from sensors to detect intruding aircraft and issue advisories for safe collision avoidance maneuvers [18]. Traditional sensors used for surveillance and tracking include ADS-B, onboard radar, and transponders [19]; however, autonomous aircraft require additional sensors both for redundancy and to replace the visual acquisition typically performed by the pilot. For this reason, vision-based traffic detection systems have been proposed, in which intruding aircraft are detected from images taken by a camera sensor mounted on the aircraft [5], [20]. In this section, we will apply our risk-driven design techniques to improve the safety of a vision-based detect and avoid (DAA) system. The code for this work can be found at https://github.com/sisl/RiskDrivenPerception.

### 4.1 Problem Setup

Figure 5 outlines the components of a vision-based DAA system. A camera mounted on the aircraft (which we refer to as the ownship) produces image observations of its surroundings. These images

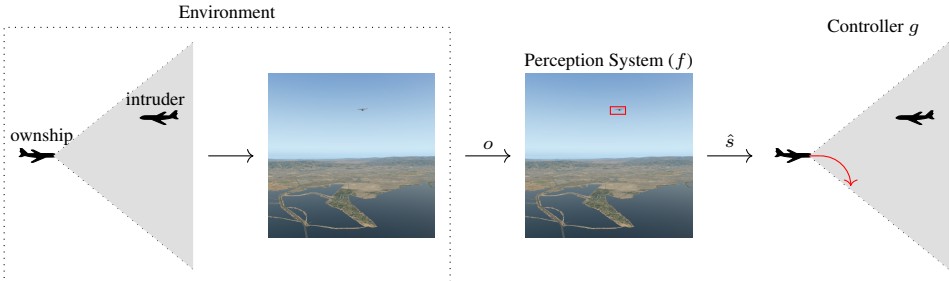

Figure 5: Overview of the vision-based detect and avoid system application.

are then passed through a perception system to produce an estimate of the state for the controller. The collision avoidance controller $g$, which is loosely based on the ACAS X family of collision avoidance systems [18], commands vertical maneuvers based on an estimate of the relative position and velocity of an intruder aircraft. Specifically, the state is defined as $s = [p, h, \dot{h}, a_{\text{prev}}, \tau]$ where $p$ is a binary variable describing whether an intruder is present, $h$ is the relative altitude of the intruder, $\dot{h}$ is the relative vertical rate of the intruder, $a_{\text{prev}}$ is the previous action, and $\tau$ is the time to loss of horizontal separation. The controller selects an action $u \in \{\text{COC}, \text{CLIMB}, \text{DESCEND}\}$ where COC represents clear of conflict. If $p = 0$, the action is always COC.

We assume that the state estimate for the controller is produced using two steps. First, the image is passed through a perception network that is responsible for detecting other aircraft (which we refer to as intruders) and producing a bounding box. Once an intruder has been detected, the ownship interrogates it to determine the rest of the state. We focus our risk-driven design efforts on the detection component of the perception system. We train a baseline network using the YOLOv5 algorithm [21], [22] on 10,000 simulated images in which the intruder location is sampled uniformly within the ownship field of view. For additional details on the controller, network architecture, computational resources, and training, see appendix B.

## 4.2 Risk-Driven Design

An object detector can make two types of errors: false positives and false negatives. While false positives have an effect on the efficiency of the system, they do not affect the risk of a collision. We therefore focus on quantifying the risk of of false negatives in the presence of an intruder. We define the perception error $\epsilon$ as zero when an intruder is properly detected and one when it is missed, leading to a perceived state

$$\hat{s} = [1 - \epsilon, h, \dot{h}, a_{\text{prev}}, \tau] \tag{10}$$

The notional error model $\pi_e(\epsilon \mid s)$ is a Bernoulli distribution with a success probability based on the detection performance of the baseline model (see appendix B for more detail). To solve for $Q_\alpha^{\pi_e}(s, \epsilon)$, we discretize the state space and apply approximate dynamic programming.

**Risk-sensitive loss function** In addition to outputting the bounding box coordinates, the detection network outputs an objectness score $\hat{p}$ representing the model's confidence that an aircraft is present in the frame. The model detects an aircraft after checking if $\hat{p}$ exceeds a threshold; however, this check poses a numerical challenge when differentiating the risk sensitive loss since the gradient is undefined at the decision threshold and is zero everywhere else. To avoid this problem we evaluate the risk of $\hat{p}$ directly by interpolating between the risk of not detecting and detecting the intruder

$$\rho_\alpha^{\pi_e}(s, \hat{p}) = (\hat{p})Q_\alpha^{\pi_e}(s, 0) + (1 - \hat{p})Q_\alpha^{\pi_e}(s, 1) \tag{11}$$

The gradient of risk function in eq. (11) with respect to $\epsilon$ is equal to the risk weight, so the model will be incentivized to be more confident in states where a detection has a significant effect on overall risk.

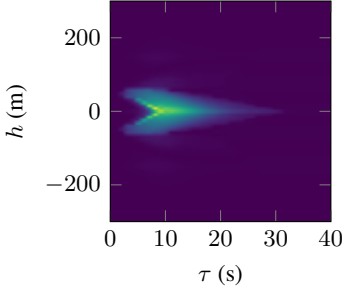

Figure 6: DAA weighting function.

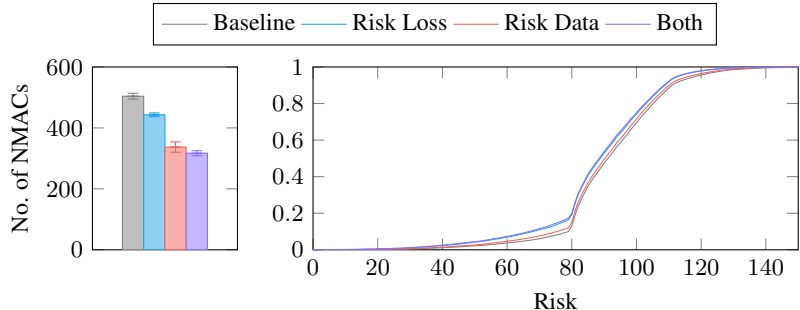

Figure 7: Number of NMACs and cumulative risk for each perception system.

**Risk-driven data generation** Figure 6 shows the weighting function when $\alpha = 0$ and an intruder is present. States with high weights correspond to states where not detecting the intruder is significantly more risky than detecting the intruder. While a collision is imminent in states with low values of $\tau$ and $|h|$, these states have low weights since a collision will occur regardless of whether the intruder is detected. The states with the highest risk weights are states in which there is just enough time to resolve an impending collision as long as the intruder is detected.

### 4.3 Evaluation

The safety of aircraft collision avoidance systems is often assessed using Monte Carlo analysis on airspace encounter models [23], [24]. Encounter models are probabilistic representations of typical aircraft behavior during a close encounter with another aircraft. To analyze the safety of a particular collision avoidance system, we can simulate the system on a set of encounters and count the number of times a near mid-air collision (NMAC) occurs. We define an NMAC as a simultaneous loss of aircraft separation to within 50 m vertically and 100 m horizontally. For the evaluation in this work, we use model consisting of 1000 pairwise encounters in which the ownship and intruder follow straight line trajectories with various relative geometries. All encounters result in an NMAC if no collision avoidance actions are taken (see appendix B.5).

We trained perception networks using four perception design techniques: no risk awareness (baseline), risk-sensitive loss function, risk-driven data generation, and both a risk-sensitive loss function and risk-driven data generation. We performed three trials for each perception design method. The first section of fig. 7 shows the number of encounters that result in an NMAC for each technique. In support of **H1**, training with the risk-sensitive loss function (blue) results in a slight increase in safety. In support of **H2**, training on data generated using the weighting function (red) results in a 33 % decrease in the number of NMACs over the baseline. With a 37 % decrease in the number of NMACs compared to the baseline, the perception systems that were developed using both risk-driven techniques (purple) are slightly safer than using either technique individually. Figure 7 also shows the cumulative distribution over the risks of the errors made by each perception system on the encounter set. This result implies that even small shifts in overall risk can have significant impacts on safety.

Figure 8 shows an example encounter that resulted in NMAC for the baseline system but was resolved by the risk-sensitive systems. The risk-sensitive system (purple) detects the intruder aircraft sooner and more often than the baseline. Moreover, the plot at the bottom of the figure shows that the risks of the perceptual errors made by the baseline system are much higher than the risks of the errors made by the risk-sensitive system.

## 5 Related Work

We discuss prior work in perception system evaluation, risk-aware control, and task-aware prediction.

**Perception system evaluation** The high-dimensional, complex nature of perception inputs makes closed-loop safety evaluation of perception systems especially challenging. This challenge was first noted in early work on computer vision, in which systems were evaluated by propagating uncertainty in a set of input images through each computer vision component [25], [26]. Similarly, a number of recent works have focused on finding adversarial perturbations that will cause misclassification

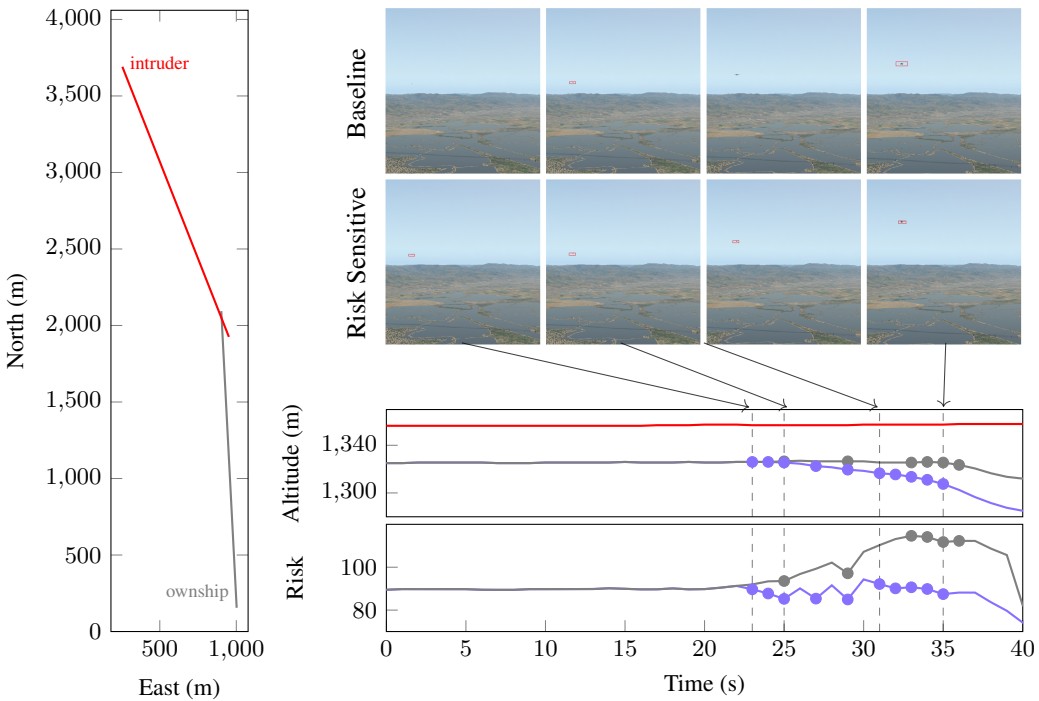

Figure 8: Example encounter that was resolved by the risk-sensitive systems but resulted in an NMAC for the baseline system. The red line denotes the intruder trajectory, while the gray and purple lines denote the ownship trajectory when using the baseline and risk-sensitive perception systems respectively. Left: overhead view of the encounter. Bottom right: altitude of each aircraft over time along with the risk of each perception error. The dotted markings indicate time steps in which the intruder was detected. Top right: perception outputs of each system at four snapshots in time.

of single input images [27]. Julian *et al.* [6], in contrast, use adaptive stress testing (AST) [28] to find sequences of adversarial perturbations that will lead a vision-based aircraft taxi system to failure rather than focusing on single images in isolation. However, this approach is only able to find failures and cannot guarantee safety if no failures are found. Katz *et al.* [29] provide approximate formal guarantees of closed-loop safety properties by using a generative model to approximate the set of plausible inputs to a perception system. Dreossi *et al.* [30] address the complexity of analyzing the closed-loop system by analyzing the controller and perception system separately before combining the results to find failures in the overall system. This work draws inspiration from this decoupled approach.

**Risk-aware control**  In safety-critical domains, it is desirable to design controllers that perform well even in worst-case conditions. This problem has been addressed through safety-constrained MDPs [31] and shielding techniques in reinforcement learning [32]. Other risk-aware control approaches use a notion of CVaR to design controllers. CVaR MDPs, for instance, have been used to produce control policies that are robust to modeling errors and worst-case outcomes [10], [14]. Distributional reinforcement learning approaches estimate the distribution of returns [15]. Using the estimate of the return distribution, it is possible to compute risk-related metrics such as CVaR and produce risk-aware reinforcement learning policies [11], [16], [17], [33]. In this work, rather than using a notion of risk to design control policies, we use it to design safe perception systems.

**Task-aware prediction**  Previous work has explored the benefit of using task-aware metrics to inform the design of predictive models. For example, Lambert *et al.* [34] note that more accurate dynamics models in reinforcement learning do not necessarily correlate with higher rewards, and Bansal *et al.* [35] show improved performance when learning a dynamics model that achieves the best control rather than learning the most accurate model. Multiple techniques have been proposed to augment traditional training loss functions with task-specific metrics to design models that are not only accurate

but also satisfy domain constraints [36]–[38]. In the context of trajectory prediction, McAllister *et al.* [39] propose control-aware prediction objectives, which take into account the downstream effects of the predictions on controller performance. These works motivate the need for task-aware objectives but are not used with image-based perception. In the context of perception, Greiffenhagen *et al.* [40] introduce the idea of connecting perception errors to overall system performance for the design and evaluation of a system that uses traditional computer vision [40]–[42]. Furthermore, Philion *et al.* [43] develop planner-centric metrics for an machine-learning based object detection system and show their benefit over traditional metrics used to evaluate object detectors. However, they focus only on the evaluation of perception systems, while this work focuses on their design.

## 6    Conclusion

In this work, we presented a methodology for risk-driven design of safety-critical perception systems. We formulated a risk function that measures the effect of perception errors on the closed-loop performance of the fully-integrated perception and control system. We then showed how to use that risk function during the design process by incorporating it into the loss function and developing a risk-driven data generation technique. We demonstrated our approach on a realistic vision-based aircraft detect and avoid system and showed that our techniques could increase safety by 37 % over a baseline system. We note that while the methods presented here focus on designing safe perception systems, they do not represent safety guarantees. Therefore, perception systems designed in this manner should still be put through additional testing and safety validation before deployment. A limitation of this work is that we assume that the perception system is Markov, which limits the applicability of our technique to perception systems that use filtering to produce state estimates. Future work will address this limitation.

## Acknowledgments and Disclosure of Funding

The NASA University Leadership Initiative (grant #80NSSC20M0163) provided funds to assist the authors with their research. This research was also supported by the National Science Foundation Graduate Research Fellowship under Grant No. DGE–1656518. Any opinion, findings, and conclusions or recommendations expressed in this material are those of the authors and do not necessarily reflect the views of any NASA entity or the National Science Foundation. This work was also supported by a grant from the UKRI Strategic Priorities Fund to the UKRI Research Node on Trustworthy Autonomous Systems Governance and Regulation (EP/V026607/1, 2020-2024).

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
