# A  Inverted Pendulum Details

## A.1  Inverted Pendulum Model and Controller

The inverted pendulum system is modeled using state $s = [\theta, \omega]$ where $\theta$ is the angle of the pendulum from the vertical, and $\omega$ is its angular velocity. The discrete-time dynamics for the inverted pendulum are represented as follows

$$\theta_{t+1} = \theta_t + \omega_t \Delta t$$
$$\omega_{t+1} = \omega_t - \frac{3g}{2\ell} \sin(\theta_t + \pi) + \frac{3a}{m\ell^2} \Delta t \tag{12}$$

where $g$ is the acceleration due to gravity, $\ell$ is the length of the pendulum, $m$ is the mass of the pendulum, $\Delta t$ is the time step, and $a$ is the input torque from the controller. In this work, we use $g = 10\,\text{m/s}$, $\ell = 1\,\text{m}$, $m = 1\,\text{kg}$, and $\Delta t = 0.05\,\text{s}$. We clip $\omega$ such that the magnitude of the angular velocity does not exceed 8 rad/s, and we clip the control inputs so that the maximum torque magnitude does not exceed 2 N·m.

We derive a simple rule-based policy to balance the pendulum according to the following two equations

$$\omega_{\text{target}} = \text{sign}(\theta)\sqrt{60(1 - \cos(\theta))}$$
$$a = -2\omega + (\omega - \omega_{\text{target}}) \tag{13}$$

where the first equation determines the angular velocity required to move the pendulum from its current angle to an angle of zero. The second equation performs proportional control using this quantity and the current angular velocity. This controller is able to keep the pendulum upright under perfect perception.

## A.2  Inverted Pendulum Risk Function

**Error model**  We model the perception errors $\epsilon = [\epsilon_\theta, \epsilon_\omega]$ as additive noise distributed according to a multivariate Gaussian distribution with zero mean and diagonal covariance. Noting that predicting $\omega$ from successive frames tends to be more difficult for a perception network than prediction $\theta$ according to our baseline model, we set the covariance as

$$\Sigma = \begin{bmatrix} 0.2 & 0.0 \\ 0.0 & 0.5 \end{bmatrix} \tag{14}$$

To solve for the risk function using distributional dynamics programming, we discretize the error model and approximate it as a categorical distribution. We select 11 discrete points for $\epsilon_\theta$ on a logarithmic scale between −0.4 and 0.4 (two standard deviations) such that the points are more highly concentrated around zero error. We determine their corresponding weights by computing their corresponding probability densities from the continuous Gaussian model and normalizing.

**State discretization**  We discretize $\theta \in [-\pi/4, \pi/4]$ and $\omega \in [-2, 2]$ using 41 points with a higher density of points around zero. To solve for the risk function, we assume an episodic setting in which each episode has 20 time steps. To reflect the episodic nature in the solving process, we add a time component to the state such that $s = [t, \theta, \omega]$ that specifies the number of steps left in the episode. For the next state in the dynamics, $t$ is decremented by one.

**Cost function**  The cost function for the inverted pendulum problem is defined as the absolute angle of the pendulum at the last time step in each episode:

$$c(s, \epsilon) = \begin{cases} |\theta|, & \text{if } t = 0 \\ 0, & \text{if } t > 0 \end{cases} \tag{15}$$

Since we specify that the pendulum fails if $|\theta| > \pi/4$, the cost is bounded between 0 and $\pi/4$. We select 50 discrete cost points on a logarithmic scale within this range for the distributional dynamic programming.

Solving for the cost distributions using dynamic programming takes under a minute on a single Intel Core i7 processor operating at 4.20 GHz. We save the distributions and use linear interpolation and

eq. (5) to evaluate the risk at an arbitrary $\epsilon$. Since the autodiff package we use for risk-sensitive training does not have a straightforward way to differentiate through an interpolation of tabular data, we train a neural network surrogate model of the risk function to use in the risk-sensitive loss. We train a small MLP network with one hidden layer containing 64 hidden units for each value of $\alpha$.

**Perception system training**  All perception systems trained for the pendulum problem were neural networks with two hidden layers with ReLU activations and 64 units each. The final layer uses a hyperbolic tangent activation (the labels are scaled to range between $-1$ and 1). We train using the ADAM optimizer with a learning rate of $1\times10^{-3}$. We train for 200 epochs for the risk data experiment and 400 epochs for the risk loss experiment.

### A.3  Inverted Pendulum Results

Table 1 shows the full results for the inverted pendulum experiments. We ran five trials for each experiment, and we report the mean and standard error of the time to failure for each experiment. Since we simulate 100 episodes of length 500 for each evaluation the highest possible mean time to failure is 500, which would indicate that the pendulum stayed upright for all time steps. The risk-sensitive design indicates a benefit for most values of $\alpha$; however, we note that performance seems to degrade when $\alpha$ becomes too large, which is likely due to being overly pessimistic. If we always assume the worst-case outcomes, there may not be any states in which a failure can be prevented. In this case, the pendulum system will not be incentivized to produce accurate estimates at any state.

Table 1: Mean time to failure for pendulum perception systems.

|  | Risk Loss | Risk Data |
| --- | --- | --- |
| Baseline | $234 \pm 62$ | $152 \pm 89$ |
| Risk Sensitive ($\alpha = 0.00$) | $\mathbf{492 \pm 6}$ | $334 \pm 102$ |
| Risk Sensitive ($\alpha = 0.20$) | $473 \pm 18$ | $\mathbf{500 \pm 0}$ |
| Risk Sensitive ($\alpha = 0.50$) | $486 \pm 12$ | $404 \pm 86$ |
| Risk Sensitive ($\alpha = 0.80$) | $489 \pm 8$ | $438 \pm 59$ |
| Risk Sensitive ($\alpha = 0.99$) | $391 \pm 59$ | $273 \pm 65$ |

## B  Vision-Based Detect and Avoid Details

### B.1  Collision Avoidance Controller

The collision avoidance controller takes in a state $s = [p, h, \dot{h}, a_{\text{prev}}, \tau]$ where $p$ is a binary variable describing whether an intruder is present, $h$ is the relative altitude of the intruder, $\dot{h}$ is the relative vertical rate of the intruder, $a_{\text{prev}}$ is the previous action, and $\tau$ is the time to loss of horizontal separation. The relative states, $h$ and $\dot{h}$, defined to be the ownship quantity minus the intruder quantity. For example, if the intruder is below the ownship, $h$ will be positive. The previous action is contained within the state to penalize undesirable operational characteristics such as reversals in collision avoidance advisory. Finally, the variable $\tau$ is used to summarize the horizontal evolution of the encounter in a single variable. The goal is to have sufficient vertical separation $h$ as $\tau$ gets close to zero to avoid a near mid-air collision (NMAC).

The controller outputs an action $u \in \{\text{COC}, \text{CLIMB}, \text{DESCEND}\}$ where COC represents clear of conflict and the CLIMB and DESCEND actions commands vertical rates. Specifically, the CLIMB action commands a vertical rate of 8 m/s, while the DESCEND action commands a vertical rate of $-8$ m/s. For COC, we let $u = 0$. Given a state $s_t$ and action $u_t$, we calculate the next state as follows

$$
\begin{aligned}
p_{t+1} &= p_t \\
h_{t+1} &= h_t + \dot{h}_t \\
\dot{h}_{t+1} &= \dot{h}_t + u_t + w \\
a_{\text{prev},t+1} &= u_t \\
\tau_{t+1} &= \tau_t - 1
\end{aligned}
\tag{16}
$$

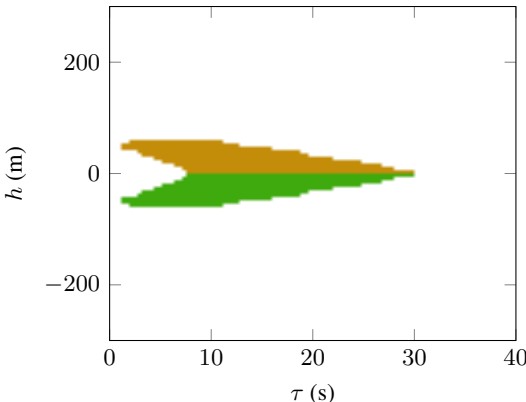

Figure 9: Policy for the collision avoidance controller for a slice of the state space where $\dot{h} = 0$ and $a_{\text{prev}} = \text{COC}$. Orange regions indicate regions of the state space where a CLIMB advisory is issued, green regions indicate DESCEND, and white areas are COC.

where $w$ represents potential noise in vertical rate due to factors such as pilot response delays. We set this noise to a categorical distribution in which $w = 0$ with 0.8 probability and $w = \pm 0.5$ with probability 0.1 each. We also provide a maximum acceleration to the model and clip $u_t$ based on the current vertical rate to comply with this maximum acceleration.

We develop a controller that is loosely inspired by the ACAS X family of collision avoidance problems by formulating the control problem as an MDP and solving for the optimal policy using dynamic programming [18]. The solving process takes under a minute on a single Intel Core i7 processor operating at 4.20 GHz. Figure 9 shows the resulting policy over a slice of the state space. As expected, the alerting region shrinks with increasing $\tau$ since there is more time to resolve collisions. No action is taken in the region near $h = 0$ and $\tau = 0$. This region represents states in which a collision is imminent and taking any action to avoid collision would be futile.

## B.2 Baseline DAA Perception Network

To train a baseline detection network, we gathered images of intruder aircraft labeled with their corresponding bounding boxes using the X-Plane 11 flight simulator, which has been used in previous work to gather data for a vision-based aircraft taxi scenario [44]. For each data point, we first sample a random ownship position and orientation in the airspace around the Palo Alto airport (PAO). We then sample an intruder position and orientation uniformly within the field of view and record the resulting image. Finally, we compute the corresponding bounding box label based on the relative position of the intruder aircraft. We train the detection networks using the YOLOv5 algorithm [21], [22] based on the open source (GNU general public license) PyTorch implementation at https://github.com/ultralytics/yolov5. For each experiment, we use the YOLOv5s model architecture, which has 7.2 million trainable parameters and train on a dataset of 10,000 images for 200 epochs with a batch size of 16 using the default hyperparameters. A single training run takes $\approx 11$ hours on a single Intel Core i7 processor operating at 4.20 GHz and adding the risk-sensitive loss component did not significantly increase the required training time.

## B.3 DAA Risk Function

**Error model**  We model the perception error for the DAA model as $\epsilon = 0$ when the intruder aircraft is detected and $\epsilon = 1$ when it is not detected. The notional error model $\pi_e(\epsilon \mid s)$ is a Bernoulli distribution with a success probability corresponding to the probability of detecting an intruder when in state $s$. We determined the parameters of the error distributions by sampling 10,000 images at states uniformly distributed throughout the state space and feeding them through the baseline (no risk awareness) perception system to determine whether the intruder is detected.

Using the results as training data, we train a small neural network with a single hidden layer of 10 hidden and a sigmoid output to represent $\pi_e(\epsilon \mid s)$. We only input part of the state to the neural

network. In particular, since $\dot{h}$, and $a_{\text{prev}}$ have no influence on the image observation and we only consider cases where $p = 1$, we omit those variables at inputs to the error model. Furthermore, noting that the error model should be roughly symmetric about $h = 0$, we input $|h|$ rather $h$. In summary, the network takes as input values for $|h|$ and $\tau$ and outputs a probability of detecting the intruder. Figure 10 shows the resulting model outputs over the state space. The cone shape is due to some states lying outside the ownship field of view.

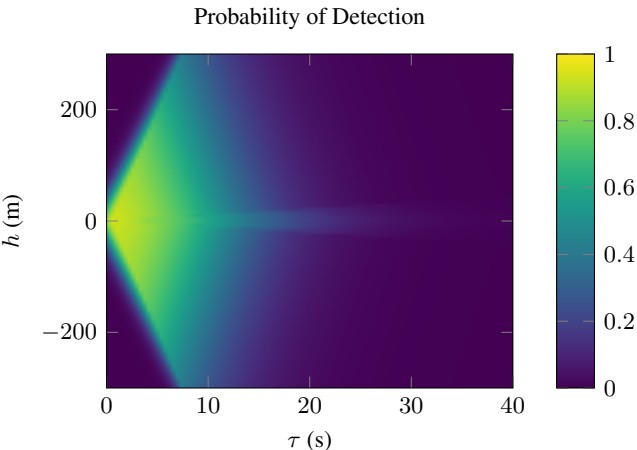

Figure 10: Perception error model for the vision-based DAA application based on the baseline model.

**State discretization** For the distributional dynamic programming, we discretize $h \in [-300, 300]$ and $\dot{h} \in [-10, 10]$ using 41 and 21 points respectively with a higher density of points around zero. We discretize $\tau \in [0, 41]$ with a uniform step size of 1. The remaining variables are already discrete.

**Cost function** We set the cost function to reflect the separation of the aircraft when $\tau = 0$. Since we want high value of cost to correspond to low separation, we set the cost function as

$$c(s, \epsilon) = \begin{cases} 150 - |h|, & \text{if } \tau = 0 \\ 0, & \text{if } \tau > 0 \end{cases} \tag{17}$$

We discretize the cost uniformly between 0 and 150 using 50 points for the distributional dynamic programming.

With these modeling assumptions, we use distributional dynamic programming to solve for $Z^\pi(s, \epsilon)$. Solving for the cost distributions takes under 10 seconds on a single Intel Core i7 processor operating at 4.20 GHz. As noted previously, $\dot{h}$, and $a_{\text{prev}}$ have no effect on the observed image at a given state, so we marginalize over these variables before calculating the risk function. We compute weights for each discrete value of $\dot{h}$ and $a_{\text{prev}}$ from simulations of the controller with perfect perception. We then take the weighted sum of the $Z^\pi(s, a)$ using these weights to obtain a cost distribution function that is only a function of $h$, $\tau$, and $\epsilon$. Using this result, we can compute the risk function.

### B.4  DAA Risk-Driven Data Generation

To generate data according the samples from the weighting function shown in fig. 6, we must map a particular $h$ and $\tau$ into a corresponding image observation. We first sample an ownship position and orientation uniformly in the airspace around the Palo Alto airport (PAO). Next, we select the intruder altitude based on the current altitude of the ownship and the current value for $h$. The remaining position variables are selected based on $\tau$; however, there is not a 1-to-1 mapping due to varying aircraft speeds and relative headings. Therefore, we sample values for the ownship speed, intruder speed, and relative heading and use these values to calculate $\tau$. Specifically, we sample speeds uniformly between 45 m/s and 55 m/s and relative headings between 120° and 240°.

## B.5 Encounter Model

**Sampling encounters** For simplicity, the encounters in this work are modeled as straight-line trajectories in which the ownship and intruder follow constant horizontal speeds. We sample an encounter by first sampling a set of encounter features according to uniform distributions over the ranges shown in table 2 and then using these features to generate trajectories for the ownship and intruder aircraft. The horizontal and vertical miss distance parameters indicate the horizontal range

Table 2: Encounter model parameters.

| Parameter | Min | Max | Unit |
|---|---|---|---|
| Ownship Horizontal Speed | 45 | 55 | m/s |
| Intruder Horizontal Speed | 45 | 55 | m/s |
| Horizontal Miss Distance | 0 | 100 | meters |
| Vertical Miss Distance | −50 | 50 | meters |
| Relative Heading | 120 | 240 | degrees |

and relative altitude of the ownship and intruder aircraft at closest point of approach. We select these distances such that all encounters will results in an NMAC if no collision avoidance action is taken. We simulate encounters that are 50 s long with the closest point of approach occurring 40 s into the encounter. The range of relative headings is set such that the encounters are close to head-on, and the intruder should almost always be within the ownship field of view (although at the beginning of the encounter it may be too small to be detected by the camera). The features in table 2 fully determine the relative trajectories of the ownship and intruder. Once the relative trajectories are generated, we rotate and shift both trajectories to put them in random positions around the Palo Alto airport.

**Simulating encounters** When we simulate the encounters, we add the perception system and ownship controller into the loop. At each time step, we read the horizontal components of the next state from the encounter. For the vertical components, we assume noisy vertical rate centered around zero until the ownship receives an alert, at which point it accelerates to the vertical rate commanded by the collision avoidance advisory.

## B.6 Additional DAA Results

In addition to simulating each vision-based DAA perception system on the encounter set, we also evaluated the precision and recall of each system on a validation set of 1000 images with uniformly sampled intruder positions within the ownship field of view. Table 3 shows the results. We note that

Table 3: Additional DAA results.

|  | Precision | Recall |
|---|---|---|
| Baseline | $0.78 \pm 0.04$ | $0.38 \pm 0.03$ |
| Risk Loss | $0.82 \pm 0.03$ | $0.37 \pm 0.02$ |
| Risk Data | $0.77 \pm 0.01$ | $0.39 \pm 0.01$ |
| Both | $0.79 \pm 0.01$ | $0.39 \pm 0.01$ |

one could improve safety of the overall system by simply biasing the perception system towards outputting a detection more often; however, this result would likely increase false positives and result in worse operational efficiency. To ensure that this effect is not causing the safety benefits we show in the risk-driven systems in fig. 7, we can analyze the effect of risk-driven training on the precision of the overall system. From these results, we determine that the risk-driven design techniques did not harm overall precision and focused their efforts only on risky states.