# OpenReview forum: "Risk-Driven Design of Perception Systems"
_NeurIPS.cc/2022/Conference — NeurIPS 2022 Accept_

### Official Review · Reviewer_nsDT · 2022-07-11

**Rating:** 6
**Confidence:** 4
**Soundness:** 2 fair
**Presentation:** 3 good
**Contribution:** 3 good

**Summary:**

The authors propose a method for quantifying the control risk induced by perception errors (state-estimation errors) in partially-observable systems (e.g. image observations).
The method assumes a dynamics model and a controller are already available and focuses on providing risk awareness to the existing system.
With a predefined probabilistic model of perception errors, risk is defined as the *Conditional Value at Risk* (CVaR) of the distribution of cumulative rollout cost (i.e. cost-to-go, or return) under the uncertainty of the assumed dynamics, control policy and perception error model.
The cost-to-go distribution is estimated using a categorical distribution, discretizing the cost space.
Once the cost-to-go density is available, the CVaR risk metric can be evaluated.
It is then used as a secondary objective for training a refined version of the perception model / state estimator, as well as for active data acquisition when training that component.
The method is evaluated on the image pendulum (simulated) and on an aircraft collision avoidance data set, demonstrating significant improvement in risk-averse behavior.


**Questions:**

See above.

**Limitations:**

I don't see any major negative societal impact. Some of the method's limitations are acknowledged in the conclusion.

**Strengths And Weaknesses:**

**Strengths**
- Accounting for estimation errors is a good idea. Blindly using bad state estimators (what's referred to as a perception system in the paper) for MPC control can have significant negative impact on performance, so the contribution appears pragmatic.
- The method appears generic and could be applied to other settings.
- The individual ablations of the active data acquisition and the risk-sensitive loss are very appreciated.
- The running pendulum example throughout the manuscript is intuitive and provides clarity w.r.t. the different aspects of the method.
- Quantifying the relative perception risk of states, by subtracting the risk associated with a perception error of 0, seems like a useful idea.

**Weaknesses**
- The method explicitly targets partially-observable systems, with an observation space (e.g. images) that differs from the Markovian state-space (e.g. joint angles) of the system.
The ideal targetted problem should be a *Partially-Observable Markov Decision Process* (POMDP) [1], whereas the authors assume an MDP for control.
A discussion of how the method deviates from the POMDP paradigm is necessary to make its limitations clear, and is currently missing.
- For proper POMDP treatment, state estimators generally need to account for the history of observations (e.g. through probabilistic filtering), or alternatively the policy can be recursive and take the history of observations into account [1]. Tying this into my previous remark, I think this needs to be mentioned in the paper; from what I understand the considered visual perception systems map only the current observation to a deterministic state (sec. 3). The authors briefly touch on this aspect in the conclusion, but I think this deserves a more detailed discussion.
- The objective for the refined state estimator is mixing state estimation and control, because the added risk-averse term still tries to maximize the CVaR of the cumulative cost under the assumed controller.
However, the refined module is still framed as a perception system / a state estimator, which dillutes the concept in my opinion.
I would suggest positioning the trained model in another way, e.g. treating it as a sub-parameterization of the overall policy which would include the controller $g$ as well.
- The method assumes that data is available in advance and can be queried freely to optimize the risk objective or perform active data acquisition. In general, this is not the case for on-line RL, where an agent would need to query the environment consecutively and data acquisition can be expensive.
- I would have preferred more systems explored in the experimental section. The method is generic, and that would support its broad applicability.

**Further remarks**
- The overview in section 3.1 could point out what will happen with the observations to prepare the reader for what's to come. Currently it only discusses quantifying the state estimation errors, but not how that fits together with the existing state estimator. Diagram in Fig. 1b is also missing the observations alltogether.
- $Z^{\pi}$ is described as a "cost distribution function" in sec. 2, but if I interpret the equations correctly it denotes the random variable of the infinite-horizon return, for which expectations, CVaR, etc. can be computed. If my interpretation is correct, I think it could be phrased better.
- I am not convinced framing the perception error model as a policy is the best idea. The original policy is still involved in the rollout in the form of $g$, the controller, and having two policies is confusing. Why not treat it as an additive error model on top of the transition, for example?


Since the method appears generic and I like the idea of quantifying the future state estimation errors of the system, I'm leaning towards a positive assessment.
However, there are conceptual issues that need to be addressed in the manuscript and I find the evaluation of the method can be improved by considering further systems.

**References**

[1] Bertsekas, D., 2012. Dynamic programming and optimal control: Volume I (Vol. 1). Athena scientific.

---

> ### Author Response · Authors · 2022-08-01
> **Response to Reviewer nsDT**
>
> We thank the reviewer for their detailed feedback on our work. You have brought up some important points which we attempt to address below.
>
> **Response to the first three comments regarding the POMDP formulation:**
>
> Deep neural networks (DNNs) have many local minima leading to a large number of equally good parameterizations for a given set of training data. So when training a DNN perception system (state estimator) there may be a variety of perception error distributions that could arise from the same set of training data.  We see the key contribution of this work as demonstrating that we can use a notion of closed-loop risk to select the DNN parameterization that makes the least risky set of perception errors while still performing just as well as the other local optima (see table 3 in appendix B.6). We chose to demonstrate this insight on a task that has a simplified controller component (one that assumes deterministic states allowing us to model the problem as an MDP), but a complex perception component, where small differences in the error distribution can lead to large differences on closed-loop safety.
>
> We completely agree that an industry-grade visual detect and avoid system should be modeled as a POMDP and involve state estimation with probabilistic filtering, and we believe that our approach can be extended to that setting. Below is a sketch that outlines a possible approach to applying risk-driven design in the context of recursive state estimation.
>
> We start by assuming that we have a control policy that depends on the current belief (possibly obtained with a POMDP solver). To account for history-dependent state estimation, we could take one of the following approaches:
> * First we would compute the risk function of making a perception error $\epsilon$ as a function of the belief $b$ rather than the state $s$. This could be done by adapting an offline POMDP solver such as point-based value iteration (Pineau et al. 2003) or taking a black-box sampling based approach (Corso et al. 2021).
> * Then, if we wanted to train a non-recurrent perception system (as in the current work), we would need to convert the belief-dependent risk function into a state-dependent one. We could do this by marginalizing over the belief space to get $\rho(s,\epsilon) =  \int_b p(s \mid b) \rho(b, \epsilon)$ which could then be used in the supervised learning context. The distribution $p(s \mid b)$ would have to be computed using the notional perception error model.
> * Alternatively, we could train a recurrent perception system that takes as input the current image as well as the belief and returns an updated belief. Under this formulation, we could directly use the belief-dependent risk function during training.
>
> These approaches are a very promising avenue for future work, but we see them as out of the scope of the current work since they require non-trivial advancencements in risk estimation algorithms (by adapting and scaling existing POMDP solving algorithms or developing black box sampling algorithms to compute CVaR in a sequential decision making system).
>
> **Comment:** The method assumes that data is available in advance and can be queried freely to optimize the risk objective or perform active data acquisition. In general, this is not the case for on-line RL, where an agent would need to query the environment consecutively and data acquisition can be expensive.
>
> **Response:** When training with the risk term in the loss function, we assume that a dataset is provided and annotated with the necessary state information. For the risk-sensitive data acquisition, we assume the capability to collect a dataset for an arbitrarily chosen set of states. We think this is a valid assumption when building modular, learning-enabled, components offline (which is currently the standard in aviation and other safety critical domains), and we don’t currently consider the online learning case.
>
> Lastly, we acknowledge and agree with the feedback under “Further remarks” and will update the manuscript to reflect these suggestions.
>
> **References:**
>
> Pineau, Joelle, Geoff Gordon, and Sebastian Thrun. "Point-based value iteration: An anytime algorithm for POMDPs." Ijcai. Vol. 3. 2003.
>
> Corso, Anthony, et al. "A survey of algorithms for black-box safety validation of cyber-physical systems." Journal of Artificial Intelligence Research 72 (2021): 377-428.

---

### Official Review · Reviewer_MFas · 2022-07-12

**Rating:** 9
**Confidence:** 4
**Soundness:** 4 excellent
**Presentation:** 4 excellent
**Contribution:** 4 excellent

**Summary:**

The paper proposes a risk driven loss and augmentation technique to improve safety performance of image based control systems. They introduce an MDP between the perception model and the controller which quantifies the risks associated with action taken at a given state, and the states which are risky. The MDP is used to quantify risk and is learned with distributional reinforcement learning. Evaluation with pendulum and a simple aircraft avoidance system shows improved results compared to vanilla perception models.

**Questions:**

I have no questions

**Limitations:**

Same as weaknesses

**Strengths And Weaknesses:**

Strengths:
- The idea is novel: use of a Q-function to assess the risk of a perception module is a departure from prior risk-sensitive systems.
- The paper is well written. The problem statement is clear, there is sufficient background provided on the method, and the implementation details make sense. Use of inverted pendulum as an example makes the paper easy to follow.
- The evaluation with an airplane avoidance system is sufficiently complex to evaluate the hypotheses posed in the paper.

Weaknesses:
- There is no details provided on how Q-learning is done. Do you use a neural network? What are the hyper-parameters? If not a neural network, what is the computational complexity?
- The paper uses discrete systems throughout. I suspect this is because of the distributional Q-learning algorithm. The paper should mention this as a limitation and perhaps discuss methods on how it can be addressed.
- The evaluation examples are simple. Evaluation on prominent benchmarks or simulators in autonomous vehicles or other realistic control systems would make the paper more convincing.
- Figure 2 is difficult to read, especially the line plots on the right.

---

> ### Author Response · Authors · 2022-08-01
> **Response to Reviewer MFas**
>
> We thank the reviewer for their helpful feedback and questions on our work. Below we address your questions:
>
> **Question:** There is no details provided on how $Q$-learning is done. Do you use a neural network? What are the hyper-parameters? If not a neural network, what is the computational complexity?
>
> **Answer:** The distributional $Q$-learning algorithm is described at a high level in the paragraph with line numbers 116-125. We used distributional dynamic programming to estimate the distribution of costs in the MDP, which is polynomial in the number of states, actions, costs, and time horizon.
>
> **Question:** The paper uses discrete systems throughout. I suspect this is because of the distributional $Q$-learning algorithm. The paper should mention this as a limitation and perhaps discuss methods on how it can be addressed.
>
> **Answer:** This statement is correct. We restricted ourselves to problems that could be discretized and solved using dynamic programming so that we could focus on our demonstration that a risk-driven approach improves our perception system on metrics of safety. We note in the aforementioned paragraph that to scale this approach to higher-dimensional systems we could employ deep distributional $Q$-learning algorithms like those developed in references 15, 16, and 17 to compute the risk function.

---

### Official Review · Reviewer_ndHK · 2022-07-16

**Rating:** 6
**Confidence:** 4
**Soundness:** 3 good
**Presentation:** 3 good
**Contribution:** 3 good

**Summary:**

The paper articulates a risk-driven approach for designing perception systems that accounts for the effect of perceptual errors on the performance of the fully integrated, closed-loop system. The key contribution is the use of conditional value at risk (CVaR) and a Markov decision process (MDP) formulation to estimate the effect of perceptual errors on the safety of a closed-loop system. A model of errors of an abstracted perception system is used as a stochastic policy of the MDP. Further, the CVaR of the distribution of future costs is used as the state-action value function leading to a policy that is based on the upper quantile of the worst-case outcomes.  The risk function computed represents the downstream impact of making a given perception error in the current state and is used in a supervised learning setting to encourage a model to avoid making high-risk errors. The risk function is also used to identify error-sensitive regions of the state space, from which additional training data can be collected. The concept is illustrated in a toy example involving pendulum dynamics and in a vision based aircraft detect and avoid application and it is shown that the collision risk is reduced by 37% over a chosen baseline.

**Questions:**

It was not clear from reading the paper if the ACAS example datasets you use are for a simulated setting or real.  How do you see the work translating to real-world settings?   How do you validate your model assumptions ?  For instance, even in this simple detection scenario, the bernoulli random variates used for probability of misdetection and probability of false alarm are complex functions of the trained yolomodel as well as the current sensing conditions (e.g. scene state, sensor noise, etc).

**Limitations:**

It is nice to see that the authors acknowledge that the concepts presented in the paper do not provide safety guarantees.  Can you comment about what specific gaps are there from taking this work to a full blown deployable system?

**Strengths And Weaknesses:**

Originality/Novelty:  The idea of systems level performance modeling and how perception errors can be linked to control for achieving particular end to end goals has been studied before. (See for instance: Greiffenhagen et al, Statistical Modeling and Performance Characterization of a Real-Time Dual Camera Surveillance System. CVPR 2000: 2335-2342.).  In this sense, the concept in the paper is well known and incremental. Other aspects including formulation of the risk sensitive loss function, dealing with differentiability issues in the loss formulation, data generation, illustration of utility in the application setting are novel.

Reference to Past work:  A number of papers have appeared in the computer vision literature in the 90’s and the 2000’s exploiting uncertainty propagation / distribution propagation for system identification. See haralick.org for several papers on performance characterization in computer vision context. The paper does not refer to any of these past literature.

Significance of work:  The work incorporates a risk sensitive loss function in the perception module design and is an illustration, in the modern ML (with supervised learning) context, of how error models in perception should be linked to overall risk in the control scenario.  The example case studies are narrow but sufficiently illustrate the concept.

---

> ### Author Response · Authors · 2022-08-01
> **Response to Reviewer ndHK**
>
> We thank the reviewer for their helpful feedback and provide responses below:
>
> **Response:** Thank you for pointing us to the suggested literature [1-3] that shows how careful modeling of a system’s components and the propagation of uncertainty can be used to evaluate or design a system that meets a set of requirements. We see three primary connections between that line of work and our present work
> 1. It introduces the idea of connecting perception errors to overall system performance for the evaluation or design of an integrated computer vision system. As you noted, our contribution is to extend this line of thinking into the supervised learning context with our risk sensitive loss and data acquisition.
>
> 2. The uncertainty propagation techniques described in prior work could be used to compute the risk function for systems that are composed of many sub components. Our setting had only two components so we could directly relate perception errors to downstream risks, but a more complex system may be better analyzed one module at a time.
>
> 3. Lastly, techniques for perception system evaluation (with a final, learned model of the perception system errors) could be used to better understand the safety of the final system. As we discuss below, our techniques do not make any guarantees on the safety of the resulting system, so additional techniques need to be applied to evaluate the system.
>
> We will integrate these references and an abridged version of this discussion into our related works.
>
>
> **Question:** It was not clear from reading the paper if the ACAS example datasets you use are for a simulated setting or real. How do you see the work translating to real-world settings?
>
> **Answer:** We generated the ACAS datasets using a simulated environment, specifically the Xplane 11 simulator. Since Xplane 11 is a photo-realistic flight simulator we have no reason to think that the safety-related improvements of the perception system wouldn’t hold with a real-world dataset. We also trained the perception system with a limited budget of 10k images, which would be a reasonable number to collect in real-world flight tests, though it may be more challenging to obtain the diversity of image data.
>
> **Question:** How do you validate your model assumptions?
>
> **Answer:** It is correct to say that mis-detection/false alarm is a complex function of the trained model and the environment. For the notional error model, we trained a neural network from data collected from a vanilla YOLO network with no risk-sensitive techniques (see Appendix B.3). The neural network captures much of the complexity of the detection error model and matches the data well according to our validation accuracy. However, this error model, no matter how accurate to the vanilla YOLO network, still won’t be representative of the error model of the yet-to-be-trained risk-driven perception system. The guaranteed mismatch between the notional error model and true error distribution of the final perception system is the main reason we use CVaR as our risk metric, since it accounts for worst-case modeling errors [4].
> So to summarize, our assumptions are that
> * We can construct an accurate model of a perception system error model from data based on the vanilla YOLO model
> * That error is not too dissimilar to the resulting error model of the YOLO model trained with our risk-driven approach (and dissimilarity can be accounted for with the appropriate choice of the CVaR parameter $\alpha$)
>
> These assumptions are ultimately validated by the fact that the risk-driven design approaches improved the safety of the overall system.
>
> **Question:** Can you comment about what specific gaps are there from taking this work to a full blown deployable system?
>
> **Answer:** There are several gaps that we hope to address in future work:
> * A deployable system will integrate a state-estimation filtering step, which requires a POMDP model. We would then have to solve our risk function in the POMDP setting which will be computationally more expensive.
> * We would need to collect and train on real-world datasets.
> * The whole system should be evaluated to better the safe operational envelope. For example, our approach does not improve performance in the presence of bad weather or nighttime conditions.
>
> **References:**
>
> [1] Greiffenhagen, Michael, et al. "Statistical modeling and performance characterization of a real-time dual camera surveillance system." CVPR 2000
>
> [2] Liu, Xufei, et al. "On the use of error propagation for statistical validation of computer vision software." IEEE transactions on pattern analysis and machine intelligence 27.10 (2005): 1603-1614.
>
> [3] Haralick, Robert M. "Performance characterization in computer vision." BMVC92. Springer, London, 1992. 1-8.
>
> [4] Chow, Yinlam, et al. "Risk-sensitive and robust decision-making: a cvar optimization approach." Advances in neural information processing systems 28 (2015).

---

> > ### Comment · Reviewer_ndHK · 2022-08-09
> > **Authors addressed review comments...**
> >
> > Dear Authors,
> > I appreciate the clear feedback.
> >
> > The model validation issue in real-world vs simulated data is a key step that is missing.
> >
> > 2) You may wish add a reference  to the Greiffenhagen et al paper since it is a longer version and addresses illustration of real-world validation more comprehensively than the CVPR 2000 paper.  A follow up paper to the 2000 paper may also be a good one to cite as it addresses evolution of design to address real-world requirements in perception.
> >
> > M. Greiffenhagen, D. Comaniciu, H. Niemann and V. Ramesh, "Design, analysis, and engineering of video monitoring systems: an approach and a case study," in Proceedings of the IEEE, vol. 89, no. 10, pp. 1498-1517, Oct. 2001, doi: 10.1109/5.959343.
> >
> > M. Greiffenhagen, V. Ramesh and H. Niemann, "The systematic design and analysis cycle of a vision system: a case study in video surveillance," Proceedings of the 2001 IEEE Computer Society Conference on Computer Vision and Pattern Recognition. CVPR 2001, 2001, pp. II-II, doi: 10.1109/CVPR.2001.991033.
> >
> > I increase my rating one level to 'weak accept'  after reading your rebuttal for all other reviews as well. The point made about experimental evaluation and illustration on general class of systems is valid.

---

### Meta-Review · Area_Chair_2zf2 · 2022-08-23

**Recommendation:** Accept
**Confidence:** Certain

**Metareview:**

This paper proposes a technique that can evaluate the risk to a system (e.g., autonomous robot, aircraft, etc.) posed by errors in its perception system. While the idea of evaluating a perception system (or any other subsystem) as part of the overall pipeline is not new, this work introduces a novel method of encapsulating the risk in the cost-to-go values of the system under assumed dynamics, control policy and perception error model. The Q-values are then learnt via distributional RL. Overall, this work has been well received and should be interesting to anyone developing complicated systems which have machine learning components in them. The authors are encouraged to incorporate all the rich feedback during author-reviewer interaction period in the camera-ready version.

**Award:**

No

---

### Decision · Program_Chairs · 2022-09-14

Accept